Vision of conspecifics decreases the effectiveness of ethanol on zebrafish behaviour

Dean Rachel 1
Hurst Radke Nicole 1
Velupillai Nirudika 2
Franczak Brian C. 2
Hamilton Trevor J. trevorjameshamilton@gmail.com 1 3
1 Department of Psychology, MacEwan University , Edmonton , Alberta , Canada
2 Department of Mathematics and Statistics, MacEwan University , Edmonton , Alberta , Canada
3 Neuroscience and Mental Health Institute, University of Alberta , Edmonton , Alberta , Canada
Johnson Sheri
Electronic publication date: 2021 Jan 15
Publication date: 2021
Volume: 9
Electronic Location ID: e10566
Received 2020 Jun 11; Accepted 2020 Nov 23
Copyright: ©2021 Dean et al.
Copyright year: 2021
Copyright holder: Dean et al.
License: This is an open access article distributed under the terms of the Creative Commons Attribution License, which permits unrestricted use, distribution, reproduction and adaptation in any medium and for any purpose provided that it is properly attributed. For attribution, the original author(s), title, publication source (PeerJ) and either DOI or URL of the article must be cited.
License URL: https://creativecommons.org/licenses/by/4.0/

Keywords: Social Buffering, Behavioural Mimicry, Ethanol, Zebrafish, Novel Object Approach Test, Boldness, Anxiety-like behaviour

Funding: Natural Science and Engineering Research Council of Canada 04853 05426 04676 This work was supported by the Natural Science and Engineering Research Council of Canada [Trevor J. Hamilton, grant number 04853, 05426; Brian C. Franczak, grant number 04676]. There was no additional external funding received for this study. The funders had no role in study design, data collection and analysis, decision to publish, or preparation of the manuscript.

==============================
Aquatic organisms in pharmacology and toxicology research are often exposed to compounds in isolation prior to physiological or behavioural testing. Recent evidence suggests that the presence of conspecifics during a stressful event can modulate behavioural outcomes (called ‘social buffering’) when testing occurs within the same context. It is unknown, however, whether the social environment during exposure interacts with the efficacy of anxiety-altering substances when subsequently tested in the absence of conspecifics. In this study, zebrafish were individually exposed to habitat water or ethanol (1.0% vol/vol) while untreated conspecifics were visually present or absent during dosing. Using the novel object approach test, a validated test of boldness and anxiety-like behaviour, we observed significantly greater effects of ethanol in isolated fish, compared to fish with a view of conspecifics during dosing. These results were not explained by altered locomotion during exposure, which might otherwise increase drug uptake. This highlights the need to consider the social environment during exposure when conducting and interpreting behavioural research involving drug or toxicant exposure.

Introduction

Living in a social environment offers many evolutionary advantages. Belonging to a group can facilitate reproduction, enable earlier detection and evasion of predators, and improve resource efficiency (Rubinstein, 1978). Social cues are commonly the mechanism that conveys these messages between conspecifics and can be used to guide responses in uncertain situations (Suboski et al., 1990). Another benefit of conspecifics is a decrease in stress level which minimizes the impact of stressful situations (Kikusui, Winslow & Mori, 2006). This phenomenon, known as ‘social buffering’, has been experimentally demonstrated in many species including cats (Masserman, 1943), goats (Liddell, 1949), rats (Davitz & Donald, 1955); (Latané, 1969), humans (Hostinar, Johnson & Gunnar, 2015), and, recently, zebrafish (Oliveira & Faustino, 2017; Faustino, Tacão Monteiro & Oliveira, 2017).

The zebrafish has become a popular model organism for use in a variety of scientific disciplines including pharmacology and toxicology. Behavioural neuroscience tests can be used to analyze a wide variety of cognitive processes in zebrafish including episodic-like memory (Hamilton et al., 2016), object recognition memory (May et al., 2016), classically conditioned memory (Sison & Gerlai, 2010), fear (Speedie & Gerlai, 2008), boldness (Dean et al., 2020), and anxiety-like behaviour (Maximino et al., 2010). To test anxiety-like behaviour there are a variety of paradigms available, with the most common being the light/dark preference and novel tank diving tests (for a review see Maximino et al., 2010). Due to the reliability of these tests and the practical simplicity in which psychopharmacological substances can be administered to zebrafish (Gerald, Lee & Blaser, 2006), adaptive behavioural responses can be easily manipulated with anxiolytic (anxiety-reducing) and anxiogenic (anxiety-enhancing) compounds (Collier & Echevarria, 2013).

Ethanol is a classic anxiolytic compound that has been shown to reduce zebrafish anxiety-like behaviour in a variety of paradigms. For instance, 0.25%, 0.5% and 1.0% ethanol increase the amount of time zebrafish spend exploring the light zone of the light/dark test (Gebauer et al., 2011; Fontana et al., 2020) and 0.3% and 0.5% ethanol increase time spent in the top zone of a novel tank dive test (Egan et al., 2009; Müller et al., 2020); both indications of the anxiolytic nature of ethanol. In the novel object approach test, used to test boldness and anxiety-like behaviour, 1.0% ethanol reduces the amount of time zebrafish spend in the outer ‘thigmotaxis zone’ adjacent to the wall of the arena (Johnson & Hamilton, 2017), further exemplifying the anxiolytic nature of ethanol. Using the same test, 1.5% ethanol also increases the time zebrafish spend in the inner zone exploring a novel object (Hamilton et al., 2017), thus increasing boldness. Taken together, these findings illustrate zebrafish display less anxiety and more boldness following an acute administration of ethanol. Recent evidence, however, suggests that the social environment in which anxiety-altering compounds are administered and/or tested in may influence the behavioural effects of these substances in zebrafish (Faustino, Tacão Monteiro & Oliveira, 2017), which may complicate conclusions.

Visual and olfactory conspecific cues can decrease the responsiveness of zebrafish to an anxiogenic compound when exposure and testing occur within the same environment (Faustino, Tacão Monteiro & Oliveira, 2017). Specifically, conspecific water and alarm substance administered next to a tank of untreated conspecifics induced significantly less freezing and erratic movements than when the adjacent tank was empty and no conspecific water was added (Faustino, Tacão Monteiro & Oliveira, 2017). When the effectiveness of each type of cue was tested, visual cues were superior to olfactory cues in reducing aversive behaviours and promoting ‘social buffering’ (Faustino, Tacão Monteiro & Oliveira, 2017). However, it is unknown whether the presence of conspecifics can alter behavioural responses when the fish are subsequently removed from the compound and tested in a separate arena. In other words, does the effect of social buffering persist beyond exposure to social cues?

In the majority of acute pharmacological experiments that test individual fish behaviour, substances are administered while fish are physically isolated, and the exposed fish is then transferred to a behavioural arena for testing (Stewart et al., 2012). In almost all studies that have examined acute ethanol exposure on behaviour of individual zebrafish, it is not clear whether conspecifics were within view during dosing (Blaser & Pénalosa, 2011; Echevarria et al., 2008; Egan et al., 2009; Fontana et al., 2020; Gerlai, Ahmad & Prajapati, 2008). Moreover, only a few studies have stated that fish were isolated during ethanol dosing (Hamilton et al., 2017; Johnson & Hamilton, 2017). To the best of our knowledge, no study has examined whether the view of conspecifics during dosing may influence the anxiety level of a fish when subsequently tested in an isolated testing arena. It is also unknown whether social buffering may also act to alter the effects of anxiety reducing, ‘anxiolytic’ substances. To explore this, we exposed individual zebrafish to either habitat water or ethanol (1.0% vol/vol) while untreated conspecifics were visually present or absent for the entire exposure period. Following exposure, the treated fish were transferred to the novel object approach test for quantification of anxiety-like behaviour and boldness (Dean et al., 2020; Krook et al., 2019; Leighton et al., 2018). Finally, we tested whether fish move at different rates and remain closer to conspecifics during the dosing period itself in order to determine whether the social condition (Isolated vs. In-view) influences behaviour during drug exposure.

Methods

Subjects and housing

Short-fin wild-type zebrafish (n = 90) were acquired from Aquatic Imports (Calgary, AB) at a minimum age of 9-months. Fish were experimentally naïve and comprised of mixed males and females (∼50/50 ratio). Following a month-long quarantine period, the fish were held in either 3 or 10L polypropylene tanks within a three-shelf bench top system (Aquatic Habitats, Aquatic Ecosystems, Inc. Apopka, FL, USA) which was controlled for filtration and aeration. No fish was ever housed in isolation and tank capacities never exceeded five fish per liter. Temperature and pH remained between 26–30 °C and 6.0–8.0, respectively. Lights were kept on a 12-hour light/dark cycle with lights on at 8 AM and off at 8 PM. Fish were fed dry brine shrimp (Omega One Freeze Dried Mysis Shrimp nutri-treat, OmegaSea Ltd., Germany) once per day, and after experimentation on test days. All experiments were approved by the MacEwan University Animal Research Ethics Board (AREB) under protocol number 05-12-13 in compliance with the Canadian Council for Animal Care (CCAC) guidelines for the care and use of experimental animals.

Experimental design

Experiment 1

This experiment used a 2 × 2 factorial design. The between-subject experimental variables included visual access to conspecifics (Figs. 1A, 1B, Isolated or In-view) and the type of substance the fish were exposed to (habitat water (CTL) or Ethanol) while in the dosing containers. Prior to experimentation, fish were randomly assigned to one of four groups: Isolated-CTL, Isolated-Ethanol, In-view-CTL, and In-view-Ethanol. Following exposure, behaviour was tested in the novel object approach test (Figs. 1C, 1D) to examine whether the social environment during exposure influences the efficacy of ethanol.

Figure 1 Experimental dosing and behavioural testing set-up.

(A) Isolated and (B) In-view dosing. An Individual fish was netted from the holding tank and placed into one of the two dosing containers. In-view fish had visual access to 12 conspecifics held in the conspecific tank but were not able to see the other fish being dosed. Fish remained in the dosing containers for 30-minutes prior to behavioural testing. (C) The circular arena used was 34 cm in diameter and 16 cm in height. (D) The novel object used was a multi-coloured LEGO figurine. E) The thigmotaxis, transition and inner zones were calibrated to 34, 23 and 12 cm in diameter respectively.

Experiment 2

A follow-up experiment was performed in which we tested the movement of individual zebrafish within the dosing containers in the Isolated or In-view conditions while exposed to control water (Figs. 2A, 2B).

Figure 2 Experimental set up for tracking behaviour during dosing.

(A) Isolated dosing, or (B) In-view dosing. The circle in the bottom left of figure (B) represents the virtual zones created in Ethovision to test whether fish spend more time on the side of the beaker closest to conspecifics when in view.

Experiment 1: Experimental conditions

Ethanol exposure

On the day of experimentation, fish were carried in their habitat tanks into the experimental room prior to feeding and were given at least 10 min to acclimatize to this new environment. A white corrugated plastic barrier was set up surrounding habitat tanks to limit external stimuli. Following the habituation period, fish were individually netted from their habitat tanks and placed into one of two experimental dosing containers (600 mL). Each dosing container contained 500 mL of solution and was also surrounded by white corrugated plastic barriers (Figs. 1A, 1B). Two dosing containers were used rather than one to increase testing efficiency by allowing two fish to be dosed with a staggered schedule. Once in the dosing container, a square piece of the same plastic was placed on top to prevent evaporation of the solution and to ensure fish remained inside (Cachat et al., 2010; Holcombe et al., 2013). Fish exposed to control water (Isolated-CTL (n = 15) or In-view-CTL (n = 15)), were placed into dosing containers that only contained habitat water (500 mL). Fish in the ethanol groups (Isolated-Ethanol (n = 15) or In-view-Ethanol (n = 15)) were placed into dosing containers with 1.0% ethanol. Solutions for each compound were made fresh each day by mixing 5.26 mL of non-denatured, 95% ethanol into 495 mL of habitat water in the respective dosing containers. The selected concentration and duration of ethanol exposure was based on previous experiments in zebrafish (Johnson & Hamilton, 2017).

View of conspecifics

Fish were assigned to either one of the two Isolated conditions (Isolated-CTL (n = 15) or Isolated-Ethanol (n = 15)) with no view of conspecifics while in the dosing container, or to one of the two In-view conditions (In-view-CTL (n = 15) or In-view-Ethanol (n = 15)) with a view of a tank containing 12 untreated conspecifics during dosing. The dosing containers used in the In-view conditions were positioned in front of each other to ensure fish in both dosing containers had equal view of their conspecifics. The same group of conspecifics were used for each In-view condition. A white corrugated plastic barrier covered the remaining two sides of the conspecific tank (Fig. 1B) and water temperatures were maintained between 26 and 30 °C by seedling heat mats (Hydrofarm Horticultural Products, Petaluma CA). Fish in the Isolated condition were surrounded fully by a white plastic barrier which was also placed on the heat mat. Fish in all conditions remained in the dosing containers for 30 min, after which the solution (including the fish) was carefully poured into a net, with a second dosing container collecting the solution. Once in the net, the fish was placed into the adjacent behavioural arena for testing. No data was acquired during dosing in experiment 1.

Experiment 1: Behavioural testing

Fish were individually tested in the novel object approach test following the 30-minute exposure period. The behavioural arena used in this experiment was circular and made from white opaque plastic (Ø= 34 cm; depth = 15 cm; Fig. 1C). The arena was placed on top of a heat mat to maintain habitat water temperatures and was surrounded by a three-sided white corrugated plastic enclosure to limit external stimuli during testing. Habitat water was added to the arena up to a maximum height of 5 cm and was replaced with fresh habitat water every four hours. An equal amount of heated habitat water was also exchanged whenever temperatures fell below 26 °C. The object used in this study was a 2 cm × 4.25 cm Lego figurine which was multi-coloured to rule out possible colour preferences (Fig. 1D; (Dean et al., 2020; Hamilton et al., 2017; Johnson & Hamilton, 2017) and was adhered using velcro to the bottom of the arena’s center. Prior to testing, three virtual zones representing the thigmotaxis, transition, and inner zones were defined using EthoVision XT motion tracking software (Fig. 1E; version 11.0, Noldus, VA, USA). All experimental procedures occurred between 9AM and 6PM prior to feeding. The time, in seconds, fish spent in each zone (thigmotaxis, transition, inner) was recorded and used as a proxy for anxiety-like behaviour (i.e., increased time in the thigmotaxis zone related to increased anxiety) and boldness behaviour (i.e., increased time in the inner zone is related to increased boldness (Ou et al., 2015). Locomotion was also assessed by tracking the distance each fish moved(cm) and their immobility(s). Fish were tested individually for a period of 10 min following dosing and recording began as soon as the fish was placed into the transition zone facing the object.

Experiment 2: Experimental conditions and behavioural testing

After the first round of experimentation we sought to determine if the social context during exposure affected the distance fish moved while in the dosing container, as this may account for the behavioural differences we observed. In a second experiment, we tested the activity of a new group of fish while in the dosing containers, tested in either the Isolated-CTL (n = 15) or In-view-CTL (n = 15) conditions (Fig. 2B). We also quantified whether fish preferred the half of the dosing container close to conspecifics (conspecific side). Following a 10-minute habituation period, one fish was individually netted from their habitat tank and placed into a 600 mL dosing container with habitat water (500 mL), identical to procedures in our first experiment. A rectangular piece of white corrugated plastic was placed beneath the dosing container to assist with motion tracking. As in the novel object approach test, a three-sided enclosure was set up during behavioural tracking and seedling heat mats maintained water temperatures. To ensure these fish received the same treatment as fish in the Isolated-CTL and In-view-CTL conditions in the first experiment, a white piece of corrugated plastic was also set up across the front of the three-sided enclosure (Fig. 2A). For fish in both the Isolated-Dosing and In-view-Dosing conditions, EthoVision was set up recorded the distance (cm) each fish moved throughout the 30-minute exposure period while in the dosing container. For fish in the In-view-Dosing condition, a habitat tank containing the same conspecifics (n = 12) that were used in the other In-view conditions was positioned to the right of the beaker. Using EthoVision, the beaker was then vertically split into two equal-sized virtual sections to compare the amount of time, in seconds, fish explored the side of the beaker closest to conspecifics (conspecific side) and the side farthest from conspecifics (empty side; Fig. 2B). To rule out external variables potentially contributing to a side preference, the habitat tank was placed to the left of the beaker for the final three of fifteen trials per condition. No differences were observed in the time spent exploring either side of the beaker regardless of whether the habitat tank was on the right or left side of the beaker (Wilcoxon Rank Sum; conspecific side, W = 10, p = 0.255) so these were combined for analysis.

Statistical analysis

R (version 4.0.2) was used to analyze the data. The majority of the utilized functions are a part of the stats library. The exceptions are the brunnermunzel.test and leveneTest functions, which are a part of the brunnermunzel and car libraries, respectively (Fox, 2016; Fox & Weisberg, 2019; Munzel & Brunner, 2000). Prior to the model fitting process, an exploratory data analysis was performed on all variables. In particular, we used the cor.test function to determine if there was evidence of statistically significant correlations among the measurements in the thigmotaxis, transitions, and inner zones and among the measurements in the distance moved and immobility variables. To analyze the effect of social condition and/or ethanol on anxiety-like behaviour and boldness, two-way ANOVAs were fitted to each variable using the aov function. The normality and constant variance assumptions were checked using Shapiro–Wilks’ and Levene’s tests, respectively, via the shapiro.test and leveneTest functions. If either assumption was violated, a Gamma Generalized Linear Model (GLM) with log link was fitted to the data using the glm function. If there was evidence that the Gamma GLM did not sufficiently fit the data, then Wilcoxon rank-sum (WRS) tests were fitted to compare both main effects and the treatment combinations of interest via the wilcox.test and pairwise.wilcox.test functions. In cases where the Gamma GLM fit sufficiently, the Brunner-Munzel test (Munzel & Brunner, 2000) was used to analyze differences between the treatment combinations of interest via the brunnermunzel.test function. Significance across all tests was determined using a 5% significance level. In what follows, we use t(df) to designate a Student’s t test statistic with df degrees of freedom, F(df1,df2) to represent a F-test statistic with df1 and df2 degrees of freedom, W to represent the test statistic from a Wilcoxon Rank Sum test, and WBM to represent the test statistic from the Brunner-Munzel test.

Results

Analysis of correlation

Time spent in the thigmotaxis zone was negatively correlated with time spent in the transition zone (r =  − 0.816, t(58) =  − 10.74, p < 0.0001) and time spent in the inner zone (r =  − 0.754, t(58) =  − 8.74, p < 0.0001). On the contrary, time spent in the inner zone was positively correlated with time spent in the transition zone, but the result was not statistically significant (r = 0.235, t(58) = 1.84, p = 0.071). Distanced moved was significantly negatively correlated with time spent immobile (r =  − 0.643, t(58) =  − 6.400, p < 0.0001). While the results show that there are significant correlations between the times spent in each zone and between distance moved and immobility we follow Hamilton et al. (2017) and Thomson et al. (2020) and study individual behavioural measures rather than consider a principal component analysis as it has been shown that these behavioural measures have a large variance across reports from different research groups.

Effect of social context

Social context was found to have a significant effect on behaviour when comparing the time fish in the In-view and Isolated conditions spent in the thigmotaxis, transition, and inner zones. Specifically, fish spent more time in the thigmotaxis zone in the In-view condition than they did in the Isolated condition (t(56) = 4.26, p <  0.0001), and less time in the transition (t(56) = 3.98, p < 0.001) and the inner zones (t(56) = 3.40, p < 0.001), on average. No significant differences were found between the mean distance fish in the In-view and Isolated conditions moved (F(1, 56) = 2.44, p = 0.124) nor between the median time fish in these groups spent immobile (W = 470, p = 0.767).

Figure 3 Effects of social context and ethanol on zone preference.

The time, in seconds, fish spent in the thigmotaxis (A), transition (B) and inner zones (C). (D) and (E) represent the effect of social context and substance on the distance fish moved (D) and the time fish spent immobile (E). Graphs show individual data points (n = 15 per group) with horizontal lines that represent mean values ± SEM. The number of * symbols identifies significant differences between groups at varying levels of significance: *P < 0.05; **P < 0.01; ***P < 0.001; ****P < 0.0001.

Investigating social context interactions

Among the control (CTL) groups, the Brunner-Munzel test provided sufficient evidence to conclude that fish in the Isolated-CTL group spent more time in the thigmotaxis zone than fish in the In-view-CTL group (WBM = − 4.34, p < 0.001, Fig. 3A). Whereas, fish in the In-view-CTL group spent more time in both the transition and inner zones than fish in the Isolated-CTL group (WBM = 3.99, p < 0.001, Fig. 3B and WBM = 4.92, p < 0.0001, Fig. 3C respectively).

Among the ethanol groups, the Brunner-Munzel test provided sufficient evidence to conclude that fish in the In-view-Ethanol group spent more time in the thigmotaxis zone than fish in the Isolated-Ethanol group (WBM = 2.07, p = 0.0497, Fig. 3A). Whereas, fish in the Isolated-Ethanol group spent more time in the inner zone compared to fish in the In-view-Ethanol group (WBM = −2.34, p = 0.030, Fig. 3C). No differences were found when comparing the time fish in the Isolated-Ethanol and In-view-Ethanol groups spent in the transition zone (WBM = − 0.93, p = 0.361, Fig. 3B).

Effect of type of solution

The type of solution was found to have a significant effect on fish behaviour. In particular, we found that fish exposed to ethanol spent less time in the in the thigmotaxis zone (t(56) = 8.78, p < 0.0001) and more time in the transition (t(56) = 7.20, p < 0.0001) and inner zones (t(56) = 9.31, p < 0.0001), on average, compared to fish dosed in control water. In addition, a significant difference was also detected between the median time spent immobile when comparing fish dosed in control water to those in ethanol (W = 169.5, p <0.0001). No significant difference was found when comparing the mean distance moved for these two groups (F (1,56) = 3.75, p = 0.058).

Investigating solution interactions

Investigating the treatment combinations revealed that time spent in the thigmotaxis zone was significantly greater for fish in the In-view-CTL group compared to fish in the In-view-Ethanol group (WBM = − 3.24, p = 0.003, Fig. 3A). Whereas, time spent in the transition zone was statistically greater for fish in the In-view-Ethanol group compared to fish in the In-view-CTL group (WBM = 3.52, p = 0.002, Fig. 3B). A significant difference was not found when comparing the time fish in these two groups spent in the inner zone (WBM = 1.82, p = 0.079, Fig. 3C). In addition, the Wilcoxon rank-sum test concluded that the median time spent immobile for fish in the In-view-Ethanol group was not significantly different compared to fish in In-view-CTL group (W = 69, p = 0.074, Fig. 3E).

Post hoc comparison of the isolated treatment groups using the Brunner-Munzel test indicated that fish in the Isolated-CTL group spent more time in the thigmotaxis zone than fish in the Isolated-Ethanol group (WBM = − 32.19, p < 0.0001, Fig. 3A). Whereas, fish in the Isolated-Ethanol group spent more time in both the transition zone (WBM = 24.80, p < 0.0001, Fig. 3B) and the inner zone (WBM = 13.16 , p < 0.0001, Fig. 3C) compared to fish in the Isolated-CTL group. The Wilcoxon rank-sum test indicated that the median time Isolated-CTL fish spent immobile was significantly greater than the median time Isolated-Ethanol fish were immobile (W = 16.5, p < 0.0001, Fig. 3E).

Distance moved and side-preference during exposure

The distance fish moved while in the dosing container did not significantly differ between Isolated-Dosing and In-view-Dosing groups (t(28) = 1.26, p = 0.220; Fig. 4A). A highly significant preference for the conspecific side of the dosing container was found in fish from the In-view-Dosing group (t(28) = 10.21, p < 0.0001; Fig. 4B).

Figure 4 Distance and side preferences during the dosing procedure.

(A) The distance, Isolated and In-view-CTL fish moved and (B) the amount of time, in seconds, In-view-CTLs spent on either side of the dosing container during dosing. The social condition did not have a significant effect on the distance fish moved, however when in-view, fish had a significant preference for the side of the dosing container closest to conspecifics. Data was analyzed using independent t-tests. Graphs show individual data points (n = 15 per group) with horizontal lines that represent mean values ± SEM. The number of * symbols identifies significant differences between groups at varying levels of significance: ****P < 0.0001.

Discussion

To examine whether the sight of conspecifics during dosing impacts behavioural measurements and the effectiveness of anxiety-altering substances, we exposed zebrafish to habitat water or ethanol (1.0%) while fish were isolated or able to observe conspecifics. Following dosing, behaviour was tested in the novel object approach test. Fish that were able to view conspecifics while being exposed to habitat (control) water had significantly less anxiety-like behaviour and increased boldness compared to fish exposed to habitat water while in isolated from conspecifics. The behavioural effects of ethanol also varied depending on the social condition in which it was administered. Ethanol had a significantly greater effect on anxiety-like behaviour and boldness in isolated fish compared to fish that were able to view conspecifics during dosing.

Both the sight of conspecifics and ethanol exposure significantly increased the time fish spent in the zones closest to the novel object (transition and inner; Figs. 3B–3C), consistent with previous research demonstrating that ethanol increases boldness (Hamilton et al., 2017; Johnson & Hamilton, 2017). Ethanol and sight of conspecifics also both decreased the time fish spent in the thigmotaxis zone; an indication that anxiety-like behaviour had decreased. The most pronounced change in behaviour occurred with the combination of ethanol exposure and isolated dosing compared to ethanol exposure and in-view dosing; anxiety-like behaviour was significantly reduced (Fig. 3A, P < 0.0001) and boldness was increased (Fig. 3C, P < 0.0001). In fish with a view of conspecifics during dosing, ethanol still decreased anxiety-like behaviour, however to much less of an extent than in isolated fish, and had little effect on approach to the novel object. A similar pattern emerged in ethanol’s influence on locomotion. Ethanol did not impact the distance Isolated or In-view fish moved (Fig. 3D), nor the time In-view fish spent immobile; it only increased immobility in Isolated fish (Fig. 3E). This suggests that social isolation either increases sensitivity to ethanol’s anxiolytic and depressant effects, or the presence of conspecifics suppresses these effects.

In an attempt to understand how the social context contributes to differences observed in behavioural outcomes, we analyzed the behaviour of a second group of fish while in the dosing container during the 30-minute dosing period. Because mobility may affect the rate of intake when fish are dosed via immersion, with greater physiological demands resulting in more ventilation and therefore increased uptake of the drug through the gills (Blaser & Vira, 2014), we sought to determine whether the heightened effect of ethanol observed in Isolated fish could be explained by greater movement during dosing. We analyzed the behaviours of fish exposed to habitat water while isolated or within view of conspecifics while in the dosing container. Interestingly, no differences were observed in the distance fish moved (Fig. 4A), indicating differences in locomotion during dosing could not explain the behaviours we observed. Not surprisingly, zebrafish spent significantly more time on the side of the dosing container closest to conspecifics when in view (Fig. 4B), demonstrating their preference to remain near other zebrafish.

An explanation for the increased anxiolytic effect of ethanol in Isolated relative to In-view groups may be related to ‘social buffering.’ Previous research in zebrafish has shown that the presence of conspecifics helps to suppress anxiety evoked by a fearful stimulus (Faustino, Tacão Monteiro & Oliveira, 2017). Faustino, Tacão Monteiro & Oliveira (2017) first demonstrated this in zebrafish by exposing fish to a conspecific alarm substance with or without the presence of conspecific cues. They found that the anxiogenic effects were dampened by the presence of olfactory and/or visual cues. In other words, fish that could observe or smell their conspecifics showed less anxiety in response to the alarm substance (Faustino, Tacão Monteiro & Oliveira, 2017). The mechanisms of social buffering have not been well explored in zebrafish; however, it is possible that zebrafish use the behaviours of their conspecifics as a source of information to guide their own responses in unfamiliar or fearful environments. This would explain why there was less of an effect of ethanol in the In-view condition in our experiment (ie. more time in the thigmotaxis zone and less time near the object) compared to the Isolated condition. Presumably, fish in the dosing container observed their conspecifics behaving normally and this may have minimized the effects of ethanol. However, social buffering has only been shown to reduce stress responses and in our study anxiolysis was decreased. The effect of ethanol may have been ‘buffered’ by the presence of conspecifics. Alternatively, the sight of conspecifics may have an anxiolytic effect on its own, as the sight of conspecifics significantly reduced the amount of time fish spent in the thigmotaxis zone, irrespective of previous ethanol exposure. However, if this were the only factor at play then the effect of ethanol should be greater in the In-view condition compared to the Isolated condition, because of the combined anxiolytic effects of conspecifics and ethanol, but this is the opposite of what we observed. Somehow the visual cues from conspecifics decreased the effectiveness of ethanol, and the candidate mechanism is social mimicry, which has been demonstrated in zebrafish (Dresoti et al., 2015). To further investigate this, future studies could manipulate the emotional state of conspecifics with stress or pharmacology and examine how these In-view fish affect the behavioural outcomes of individual fish observing them. In this study, we did not examine sex differences, manipulate the number of fish used as conspecifics, nor manipulate the distance of the conspecific tank from the focal fish. These factors may also impact the strength of the social buffering effect. It would also be valuable to explore the neurochemical basis of social buffering with analysis of brain chemistry after dosing in these social conditions.

Conclusions

Taken together, the sight of conspecifics decreases anxiety, and more so, buffers the anxiolytic effect of ethanol. These findings have important implications in the fields of pharmacology, toxicology and behavioural neuroscience as isolated drug administration seems to be more effective in eliciting a behavioural response. Additionally, zebrafish behavioural research findings can be inconsistent across laboratories, and this may be due a lack of detailed methodological reporting, including whether conspecifics are within visual range during drug exposure. Social buffering may offer a potential explanation for at least some of these discrepancies and necessitates detailed descriptions of dosing conditions in future experiments. Overall, this study provides the first evidence that the social condition during dosing effects the efficacy of anxiolytic substances when subsequently tested in isolation. This highlights the need to consider the social environment during exposure when conducting or interpreting behavioural research.

Supplemental Information

Supplemental Information 1 Raw data

Click here for additional data file.

We would like to thank Jasmin Bajwa, Shayna Chaput, Dr. Melike Schalomon, and Aleah McCory (Animal Care Technician) for their help with daily husbandry and aquarium maintenance.

Additional Information and Declarations

Competing Interests

Author Contributions

Animal Ethics

Data Availability

The authors declare there are no competing interests.

Rachel Dean conceived and designed the experiments, performed the experiments, analyzed the data, prepared figures and/or tables, authored and reviewed drafts of the paper, and approved the final draft.

Nicole Hurst Radke analyzed the data, authored and reviewed drafts of the paper, and approved the final draft.

Nirudika Velupillai and Brian C. Franczak analyzed the data, prepared figures and/or tables, authored and reviewed drafts of the paper, and approved the final draft.

Brian C. Franczak analyzed the data, prepared figures and/or tables, authored and reviewed drafts of the paper, and approved the final draft.

Trevor J. Hamilton conceived and designed the experiments, analyzed the data, prepared figures and/or tables, authored and reviewed drafts of the paper, and approved the final draft.

The following information was supplied relating to ethical approvals (i.e., approving body and any reference numbers):

All experiments were approved by the MacEwan University Animal Research Ethics Board (AREB) under protocol number 05–12–13, in compliance with the Canadian Council for Animal Care (CCAC) guidelines for the care and use of experimental animals. The authors confirm that all experiments were performed in accordance with these guidelines and regulations.

The following information was supplied regarding data availability:

The raw data is available as a Supplemental File.

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
