# Peer review of "Vision of conspecifics decreases the effectiveness of ethanol on zebrafish behaviour"

_PeerJ, doi:10.7717/peerj.10566_

## Round 0.1 · original submission · Major Revisions

This study is well written and certainly novel, but the statistical analyses, reporting, and interpretation of the data are not adequate. It is not clear which tests were parametric and which were non-parametric – I suspect it is the time data that does not meet normality? Unfortunately, the use of the non-parametric analyses does not match your experimental design, and you present the data as if you performed three separate experiments:

1. control isolated vs. control in-view
2. isolated ethanol vs. isolated control
3. in-view ethanol vs. in-view control

To claim that the presence of conspecifics lessens the effects of ethanol, you need to report on the interaction. I presume the reason why this is missing is because of the non-parametric statistics. Have you attempted to apply a transformation on the data, either log or square root? Alternatively, as suggested by reviewer 1, you could use a linear model rather than a non-parametric test, as you can use a gamma distribution for latency or timing data. I would have no issue with non-parametric tests if you had only one categorical variable, but in order to meet your experimental design objectives, something other than non-parametric tests will need to be done. There also needs to be some justification for why males and females were not compared in the analysis, as the sexes can vary greatly in their behaviour. I suspect there is not enough power to include sex in your models, so the choice to not compare sexes needs to be addressed in the discussion. I would suggest that the authors present plots or means/standard errors of both sexes, as supplementary material. Finally, test-statistics and degrees of freedom need to be presented throughout, not just p-values.

I note that reviewer 1 queries why you used a separate set of fish to look at distance moved and side preference. I assume you did this as you did not film the exposure beakers during the original experiment, so this second experiment is an afterthought. Hence, I do not have the same concerns, except that you always put conspecifics on the same side, meaning any bias from lighting, etc. may have confounded your findings.

I also note that reviewer 1 has concerns regarding the presentation of data for all three zones. I suggest that the authors report the correlation between the variables, and then justify the presentation of all three, including citing recent publications from this lab group. Using a principal component analysis is an alternative approach to analyse these data.

I agree that Table 1 does not add to the manuscript, in its current form. The table should be included as supplementary material, and some discussion of the overall trends/patterns included in the main text – see reviewer 1 comments on this.

Please see some additional suggested edits in the annotated pdf.

Final acceptance of your manuscript will depend on a satisfactory resolution of the issues raised by the referees. When sending a revised version of your paper, please explain how you dealt with each of the points raised by the referees and myself.

Reviewer 1 ·

Basic reporting

The introduction is lacking background and references on social isolation/social environment on zebrafish anxiety response

The known anxiolytic effect of ethanol on zebrafish is not mention in the introduction and not until part way through the methods on line 137. It should be mentioned earlier in the manuscript

L.84-85: You start the sentence by saying : "recent evidence..." but failed to cite any references. Please add references at the end of the sentence

L.87-96: in this paragraph you mixing 2 concepts: That visual and olfactory conspecific cues play a role in reducing stress/fear in zebrafish AND that exposing and testing fish in the same environment play a role in reducing stress/fear response. Maybe try to make the distinction clearer by having 2 separates paragraphs

Not sure Table 1 is needed since the authors don't analyze or discuss the data in details. The authors could either discuss the table in more details or (my preferred option) get rid of the table and just have a sentence saying half of the studies looking at the effect of ethanol om...published the last ... years had conspecifics in view….. Also the table is very small and hard to read

Fig 3 needs a legend that explains the colour coding

In your reference list, make sure the species Latin names are italicized.

Experimental design

The methods are well described but could be clearer in some places:
- lines 122 :fish were never housed in isolation. For that to be relevant, we need more background on the effect of social isolation on fish (see earlier comment about the introduction)
- lines 161-162: why not testing the fish in the same arena that it has received ethanol?
- it is not clear why you are using another group of fish to look at distance moved and side preference (it is briefly mentioned in the discussion on l 278-280, this should appear here, in the methods). Presumably, you could have measured those variables using Ethovision while also measuring time in thigmotaxis. time in inner zones...Ethovosion allows you to have multiple zones on top of each other. Testing another group adds more variables that hard to control for (group, time effect...)

Line 214: 3 trials out of how many? Please specify

Line 223: I would recommend using a parametric test with a non-gaussian assumption (e.g.: glm) and instead of running 3 different ANOVA, having all the variables in the same model.

Also, since you used a mix of males and females. have you look at the effect of sex on boldness behaviour? You should add the variable sex in your analyses. See Roy and Bhat (2018) or Dahlbom et al 2011 for sex difference in boldness behavior in zebrafish

Validity of the findings

Statistics: You must provide F values and df when presenting ANOVA results.
Have you check for multicollinearity? Presumably variables such as thigmotaxis, transition and inner zones will be highly correlated variables. It may pay to have only 2 out of those 3 variables in the model rather than having the 3 as it may cause the SD to be inflated. Or alternatively, you could present only 1 of the 3 variables you looked at since presenting the 3 is redundant (i.e.: if a fish spend 70% of its time on the outer zone, it won’t spend more than 30% in the other 2, and knowing only the time spent in the outer zone OR inner zone might be enough)


Conclusion: you results show that the presence of conspecifics decreased boldness OR increased anxiety OR 'canceled' the anxiolytic effect of ethanol. From your results it is not possible to tell which conclusion best fits your results. Something you failed to discussed was the side preference. You always put the conspecifics on the right side of the tank when you tested. It would have been better to randomly placed them on the right or left side to control for that preference. Also, since the focal fish seemed to be attracted by the conspecifics, they would have likely spent more time in the thigmotaxis zone, close to the other fish, influencing greatly the way you measured boldness

Also, something that should be mention is zebrafish visual acuity. How far can zebrafish see? You could look at the literature on the topic and give recommendations regarding the minimum distance fish tanks should be spaced apart when testing fish, to avoid social buffering

Additional comments

Dear authors
Your research question is original and very relevant to pharmacological and toxicological studies involving zebrafish
However, the statistics were not adequate, the methods lacked clarity in places and the discussion is missing some important points (see comments above)

Reviewer 2 ·

Basic reporting

The writing is clear and unambiguous, with appropriate use of references and article structure.

Experimental design

The experiments are designed appropriately and appear to have been carried out in a rigorous manner.

Validity of the findings

The conclusions are clearly stated and well justified by the data presented.

Additional comments

This manuscript details a set of straightforward and seemingly well-executed experiments. The text is clear and easy to follow. As such, I have only a few very minor suggested changes, after which I recommend publication in PeerJ:

Title: I would consider expanding the title to be more easily interpretable / usable by potential readers. I understand that short titles can also help to attract readers but I feel that the current title is too broad. It could potentially be good to mention something about social environment during exposure? Also, instead of “effectiveness of ethanol”, maybe you could be more specific and mention that you investigated the anxiolytic effects of ethanol? My rough attempt at a title like this: “Social context during exposure mediates the anxiolytic effects of ethanol in zebrafish”. This is a rough attempt only so I am happy for the authors to take or leave my suggestions, although I do recommend a more detailed title generally.

Line 63: Perhaps “Belonging to a group can facilitate …”? Not all points mentioned in the list of advantages of social living are true for all social-living species.

Line 70: There should be a comma before “and recently,”

Line 80: Perhaps “can be administered to zebrafish”?

I also have a few more general comments that might be useful / relevant to the authors’ upcoming studies:

1) I recommend that, for future studies, the authors consider using multiple exposure concentrations so that a dose-response curve can be generated. Furthermore, it would be very interesting to see if changes in the number of conspecifics in view throughout the exposure period might influence this finding (i.e. maybe the group size of the conspecifics that were in view is important).
2) The sample sizes (15 fish per treatment) are quite low for a behavioural study of this nature, given that behaviour is inherently variable and individuals are likely to be differentially affected by both ethanol exposure and conspecific presence / absence. Was statistical power analysis carried out prior to this study to ensure that significant effects (if any) would be detected with sufficient sensitivity? I suggest that the authors explore power analysis for future similar studies and that 20 individuals (ideally more than 25) be used in treatments such as these—although this should be decided on a case-by-case basis and will, of course, also depend on ethical considerations to do with the number of animals being used.

---

## Round 0.2 · Minor Revisions

I thank the authors for their detailed response to reviewer and editor comments. Most of the comments and queries have been dealt with, but you’ll see that Reviewer 1 still has some concerns, and I detail some additional suggestions as well, to help improve the manuscript.

The presentation of the results:
I think the issue is the order in which you present the results. You have a 2x2 model, so the general method is to report on the two main effects (the effect of the solution type and the effect of social environment), then to report on the interaction effect (are these actually ever reported, or only the posthoc tests?), and then to break it down into the posthoc comparisons. I think it really just requires re-ordering how you present the results, and including the statistics for the interaction, if not already there?

I also wonder why you bothered to use a 2-way ANOVA for distance, why not just use a standard glm, as the data are normally distributed, yes?

Note that you should also include the packages and functions used in R and you provide the version of R, not Rstudio – there’s a difference, it’s the version of R that is important to report.

The first paragraph of the results should detail the direction of the effect, rather than just stating that social context had a significant effect.

Methods:
I agree with the reviewer that the methods section is not presented as clearly as it could be. Your experimental design section contains information (lines 163-173) that is more relevant to your dosing exposure and behavioural phenotyping? Would it not make more sense to follow the experimental design section with a section on Exposure Regime (including the second experiment to assess side preference) and then a section on Behavioural Phenotyping?

Correlation analysis/PCA recommendation:
As for the comments about correlations. I don’t feel that a PCA is necessary, as I personally prefer to report on individual traits as well, but I do feel that you need to report correlations, using all 60 datapoints together. I don’t know why you would break the data down into the four groups and analyse correlations that way, as the data are analysed all together in your models. As suggested before, report the correlations between the variables, but explain that you prefer to report on each variable, and justify why – in the main text, not only in the response.

Supplementary table:
I also agree that the table is too small to read. I also I think you are missing a key column of information – you have a ‘conspecifics in view during exposure’ column, but there is a very big difference between conspecifics in view because they were dosed together, vs. a tank of conspecifics nearby. I think you need to clarify this issue in the main text as you currently state that it is common for aquatic species to be exposed in isolation, but it’s actually a lot easier to dose in groups, so how common is this? I suggest you include this information in the table as well. I believe there are also some errors in the table. For example, the Dlugos and Rabin study exposed fish in 700 mL, not a 10L tank. The Gerlai et al 2000 study (note listed as 2001 in the reference list) exposed fish as a group in a 30x15x10 tank, whereas the current table lists the dosing container as dimensions unspecified. I didn’t check all studies, I was just trying to figure out whether the table also included studies that dosed in groups and came across these errors. The supplementary table should have it’s own reference list – unless every study that is presented in the table is also reference in the main text?


A few other comments:
Sometimes you focus only on boldness behaviour, other times only on anxiety, but then you also assessed exploratory behaviour, so I think you need to be clear, throughout, about what behaviours you were measuring and which response variables represent which behaviours.

I suggest you include something about behaviour in your title – otherwise, your paper could be about anything- gene expression, neurotransmitters, mating behaviour, etc. - ultimately your decision though.

Reference to Fig 4 first appears in the discussion – shouldn’t you be reporting the data in the results?

I am not concerned with reviewer 1s comment about why you didn’t film fish during your main experiment, as most people wouldn’t film dosing, so no response needed for this comment.

It’s not ideal that you only randomized 3 of the 15 trials, but you at least provide the detail, and do a statistical comparison, so no further response required.

Reviewer 1 ·

Basic reporting

The authors did a good job at revising their manuscript. Whereas the introduction and discussion read a lot better, the methods are still confusing and I'm not convinced by the statistical results.
Table 1, even though it has been moved in the appendix is still too small to be readable.

Experimental design

The methods should be streamlined, for example
1-subject and housing
2- dosing (ie.: explanation of the treatments, not filmed, no data collected)
3- anxiety test (explanation of the test and data collected)
4- activity test (explanation of the test and data collected)
At the moment, there a lot of repetitions which makes your methods unclear. Also separating the ethanol/water and in view/isolated treatments in 2 paragraphs is not helping, it should all be combined in 1 paragraph
Also, I'm not convinced with your explanation as to why you did not film fish activity during the dosing. Your experimental design in both Fig 1B and 2B look similar. you should have been able to film both
L. 221-223 unclear, not sure what you mean. please rephrase
L. 241-242: so you did randomize the side the tank was placed but only 3 out of 15 trials. Why? why not do half on one side and half on the other
L.245: p value is repeated twice. Please report U and p value

Validity of the findings

I cannot see statistics reporting interactions in your manuscript. I'm not sure what the W reported l. 290-306 is referring to. Looks like you are doing 2x2 comparisons, which would be alright to present if you had significant interaction in the first place.
Regarding correlations, some variables are definitively correlated. Therefore I still recommend using PCA analyses for your data and used that scored to look at the effect of your treatments on anxiety. See for example: Magnhagen, Carin, and Jost Borcherding. "Risk-taking behaviour in foraging perch: does predation pressure influence age-specific boldness?." Animal Behaviour 75.2 (2008): 509-517.

---

## Round 0.3 · Minor Revisions

Thank you for your thorough and thoughtful responses to review comments. I think the manuscript it looking great, there's just the issue of the correlation aspect. The differences between your treatment combinations are analysed using the entire dataset, and you would do the same with a PCA. Basically, you would perform a PCA with your three zone responses, and then the component explaining most of the variation would be used as your response variable to test for differences between the treatment combinations. As I have previously indicated, I don’t believe you need this. Nor do you need the table you have provided, which isn’t very informative without p-values anyways. All I was really asking for is a simple acknowledgement that while data may be correlated between the three zones and also between distance moved and immobility variables (no need to look at the correlation between the latter two variables with the zone variables), that you prefer to report on each individual variable. I’ll provide an example from a recent paper published by my group:

“Distance travelled was negatively correlated
with bottom time (r = −0.39, df = 203, p < .0001), stationary time (r
= −0.89, df = 203, p < .0001) and exploration (r = −0.61, df = 178;
p < .0001). In contrast, bottom time was positively correlated with
stationary time (r = 0.46, df = 203, p < .0001) and exploration
(r = 0.66, df = 178, p < .0001), and stationary time was positively
correlated with exploration (r = 0.65, df = 178, p < .0001). Despite
these correlations, we present data for individual behavioural measures, rather than take a principal component approach, as behaviours measured in studies tend to be highly variable across research groups.” Thomson et al., 2020. Long-term repeatability of behaviours in zebrafish. Ethology.

In your situation, you can also refer to recent publications, like your 2017 Scientific Reports paper, where you present all three variables.

Note that you report your Hamilton et al., 2017 pub as a 2017b, but there is no 2017a in the reference list?

I also provide minor edits and suggestions in the attached pdf (see red highlighted edits in the pdf).

---

## Round 0.4 · accepted · Accept

Thank you for revising the manuscript.